# Model-Agnostic Round-Optimal Federated Learning via Knowledge Transfer

## Abstract

Federated learning enables multiple parties to collaboratively learn a model without exchanging their local data. Currently, federated averaging (FedAvg) is the most widely used federated learning algorithm. However, FedAvg or its variants have obvious shortcomings. It can only be used to learn differentiable models and needs many communication rounds to converge. In this paper, we propose a novel federated learning algorithm FedKT that needs only a single communication round (i.e., round-optimal). With applying the knowledge transfer approach, our algorithm can be applied to any classification model. Moreover, we develop the differentially private versions of FedKT and theoretically analyze the privacy loss. The experiments show that our method can achieve close or better accuracy compared with the other state-of-the-art federated learning algorithms.

## 1 Introduction

While the size of training data can influence the machine learning model quality a lot, the data are often dispersed over different parties in reality. Due to regulations on data privacy, the data cannot be centralized to a single party for training. To address these issues, federated learning (Kairouz et al., 2019; Li et al., 2019a;b; Yang et al., 2019) enables multiple parties to collaboratively learn a model without exchanging their local data. It has become a hot research topic and shown promising results in the real world (Bonawitz et al., 2019; Hard et al., 2018; Li et al., 2020a; Peng et al., 2020).

Currently, federated averaging (FedAvg) (McMahan et al., 2016) is a widely used federated learning algorithm. Its training is an iterative process with four steps in each iteration. First, the server sends the global model to the selected parties. Second, each of the selected parties updates its model with their local data. Third, the updated models are sent to the server. Last, the server averages all the received models to update the global model. There are also many variants of FedAvg (Li et al., 2020c; Karimireddy et al., 2020). For example, to handle the heterogeneous data setting, FedProx (Li et al., 2020c) introduces an additional proximal term to limit the local updates, while SCAFFOLD (Karimireddy et al., 2020) introduces control variates to correct the local updates. The overall frameworks of these studies are still similar to FedAvg.

FedAvg or its variants have the following limitations. First, they rely on the gradient descent for optimization. Thus, they cannot be applied to train non-differentiable models such as decision trees in the federated setting. Second, the algorithm usually needs many communication rounds to finally achieve a good model, which causes massive communication traffic and fault tolerance requirements among rounds. Last, FedAvg is originally designed for the *cross-device* setting (Kairouz et al., 2019), where the parties are mobile devices and the number of parties is large. In the *cross-silo* setting where the parties are organizations or data centers and the number of parties is relatively small, it is possible to take better advantage of the computation resources of the parties with relatively high computation power.

In order to address the above-mentioned limitations, we propose a novel federated learning algorithm called FedKT (Federated learning via Knowledge Transfer) **focusing on the cross-silo setting**. With the round-optimal design goal, FedKT extends the idea of ensemble learning in a novel 2-tier design to federated setting. Inspired by the success of the usage of unlabelled public data in many studies (Papernot et al., 2017; 2018; Jordon et al., 2019; Chang et al., 2019), which often exists such as text and images, we adopt the knowledge transfer method to reduce the inference and storage costs of ensemble learning. As such, FedKT is able to learn any classification model including differentiable

models and non-differentiable models. Moreover, we develop differentially private versions and theoretically analyze the privacy loss of FedKT in order to provide different differential privacy guarantees. Our experiments on four tasks show that FedKT has quite good performance compared with the other state-of-the-art algorithms.

Our main contributions are as follows.

- We propose a new federated learning algorithm named FedKT. To the best of our knowledge, FedKT is the first algorithm which does not have any limitations on the model architecture and needs only a single communication round.
- We show that FedKT is easy to achieve both example-level and party-level differential privacy and theoretically analyze the bound of its privacy cost.
- We conduct experiments on various models and tasks and show that FedKT can achieve comparable accuracy compared with the other iterative algorithms. Moreover, FedKT can be used as an initialization step to achieve a better accuracy combined with the other approaches.

## 2 BACKGROUND AND RELATED WORK

### 2.1 ENSEMBLE LEARNING

Instead of using a single model for prediction, ensemble learning (Zhang & Ma, 2012) combines the predictions of multiple models to obtain better predictive performance. There are many widely used ensemble learning algorithms such as boosting (Rätsch et al., 2001) and bagging (Prasad et al., 2006). One important factor in ensemble learning is the model diversity. The increased model diversity can usually improve the performance of the ensemble learning. In federated learning, since different parties have their own local data, there is natural diversity among the local models. Thus, the local models can be used as an ensemble for prediction. Previous works (Yurochkin et al., 2019; Guha et al., 2019) have studied ensemble learning for federated learning and demonstrated promising predictive accuracy. As mentioned in their studies, since the prediction involves all the local models, the inference and the storage costs are prohibitively high especially when the number of models is large. In our study, we also use the local models as an ensemble and further use knowledge transfer to learn a single model in order to reduce the inference and the storage costs.

### 2.2 KNOWLEDGE TRANSFER OF THE TEACHER ENSEMBLE

Knowledge transfer has been successfully used in previous studies (Hinton et al., 2015; Papernot et al., 2017; 2018; Jordon et al., 2019). Through knowledge transfer, an ensemble of models can be compressed into a single model. A typical example is the PATE (Private Aggregation of Teacher Ensembles) (Papernot et al., 2017) framework. In this framework, PATE first divides the original dataset into multiple disjoint subsets. A teacher model is trained separately on each subset. Then, the max voting method is used to make predictions on the public unlabelled datasets with the teacher ensemble, i.e., choosing the majority class among the teachers as the label. Last, a student model is trained on the public dataset. A good feature of PATE is that it can easily satisfy differential privacy guarantees by adding noises to the vote counts. Moreover, PATE can be applied to any classification model regardless of the training algorithm. PATE is not designed for federated learning. Inspired by PATE, we propose FedKT, which adopts the knowledge transfer approach in the federated setting to address the limitations of FedAvg.

### 2.3 FEDERATED LEARNING WITH A SINGLE COMMUNICATION ROUND

There are several preliminary studies on federated learning algorithms with a single communication round. Guha et al. (2019) propose an one-shot federated learning algorithm to train support vector machines (SVMs) in both supervised and semi-supervised settings. Instead of simply averaging all the model weights in FedAvg, Yurochkin et al. (2019) propose PFNM by adopting a Bayesian nonparametric model to aggregate the local models when they are multilayer perceptrons (MLPs). Their method shows a good performance in a single communication round and can also be applied in multiple communication rounds. While the above two methods are designed for specific models

(i.e., SVMs in Guha et al. (2019) and MLPs in Yurochkin et al. (2019)), we propose a general federated learning framework which is applicable to any classification model.

## 2.4 FEDERATED LEARNING WITH KNOWLEDGE TRANSFER

There are some related studies (Li & Wang, 2019; Chang et al., 2019) using knowledge transfer in federated learning. However, Li & Wang (2019) has a different setting with us while Chang et al. (2019) has a different objective with us. In (Li & Wang, 2019), a public labeled dataset is needed to conduct initial transfer learning, while FedKT only needs a public unlabelled dataset. Chang et al. (2019) designs a robust federated learning algorithm to protect against poisoning attacks. The performance of their approach is slightly worse than FedAvg. We notice that there are some recently published contemporaneous work (He et al., 2020; Lin et al., 2020). He et al. (2020) considers cross-device setting and uses group knowledge transfer to reduce the overload of each edge device. Lin et al. (2020) utilizes knowledge transfer only in the server side to further update the averaged global model.

All existing approaches transfer the prediction vectors (i.g., logits) on the public dataset between clients and the server. FedKT transfers the voting counts and thus can easily satisfy differential privacy guarantees with a tight theoretical bound on the privacy loss. Moreover, FedKT is designed with a round-optimal goal, while the other approaches use iterative learning algorithms that needs many communication rounds to converge.

## 2.5 DIFFERENTIAL PRIVACY

Differential privacy (Dwork, 2011; Dwork et al., 2014) is a popular standard of privacy protection. It guarantees that the probability of producing a given output does not depend much on whether a particular data record is included in the input dataset or not. It has been widely used to protect the machine learning models (Shokri & Shmatikov, 2015; Abadi et al., 2016; Li et al., 2020b).

**Definition 1.** $((\varepsilon, \delta)$-Differential Privacy) Let $\mathcal{M} \colon \mathcal{D} \to \mathcal{R}$ be a randomized mechanism with domain $\mathcal{D}$ and range $R$. $M$ satisifes $(\epsilon, \delta)$-differential privacy if for any two adjacent inputs $d, d' \in \mathcal{D}$ and any subset of outputs $S \subseteq \mathcal{R}$ it holds that:

$$\Pr[\mathcal{M}(d) \in S] \le e^\epsilon \Pr[\mathcal{M}(d') \in S] + \delta. \tag{1}$$

The moments accountant method (Abadi et al., 2016) is a state-of-the-art approach to track the privacy loss. We briefly introduce the key concept, and refer readers to the previous paper (Abadi et al., 2016) for more details.

**Definition 2.** (Privacy Loss) Let $\mathcal{M} \colon \mathcal{D} \to \mathcal{R}$ be a randomized mechanism. Let $\mathsf{aux}$ denote an auxiliary input. For two adjacent inputs $d, d' \in \mathcal{D}$, an outcome $o \in \mathcal{R}$, the privacy loss at $o$ is defined as:

$$c(o; \mathcal{M}, \mathsf{aux}, d, d') \triangleq \log \frac{\Pr[\mathcal{M}(\mathsf{aux}, d) = o]}{\Pr[\mathcal{M}(\mathsf{aux}, d') = o]}. \tag{2}$$

**Definition 3.** (Moments Accountant) Let $\mathcal{M} \colon \mathcal{D} \to \mathcal{R}$ be a randomized mechanism. Let $\mathsf{aux}$ denote an auxiliary input. For two adjacent inputs $d, d'$, the moments accountant is defined as:

$$\alpha_{\mathcal{M}}(\lambda) \triangleq \max_{\mathsf{aux}, d, d'} \alpha_{\mathcal{M}}(\lambda; \mathsf{aux}, d, d') \tag{3}$$

where $\alpha_{\mathcal{M}}(\lambda; \mathsf{aux}, d, d') \triangleq \log \mathbb{E}_o[\exp(\lambda c(o; \mathcal{M}, \mathsf{aux}, d, d'))]$ is the log of moment generating function.

The moments have good composability and can be easily converted to $(\varepsilon, \delta)$-differential privacy Abadi et al. (2016).

**Party-level Differential Privacy** In addition to the standard example-level differential privacy, party-level differential privacy (Geyer et al., 2017; McMahan et al., 2018) is more strict and attractive in the federated setting. Instead of aiming to protect a single record, party-level differential privacy ensures that the model does not reveal whether a party participated in federated learning or not.

**Definition 4.** (Party-adjacent Datasets) Let $d, d'$ be two datasets of training examples, where each example is associated with a party. Then, $d$ and $d'$ are party-adjacent if $d'$ can be formed by changing the examples associated with a single party from $d$.

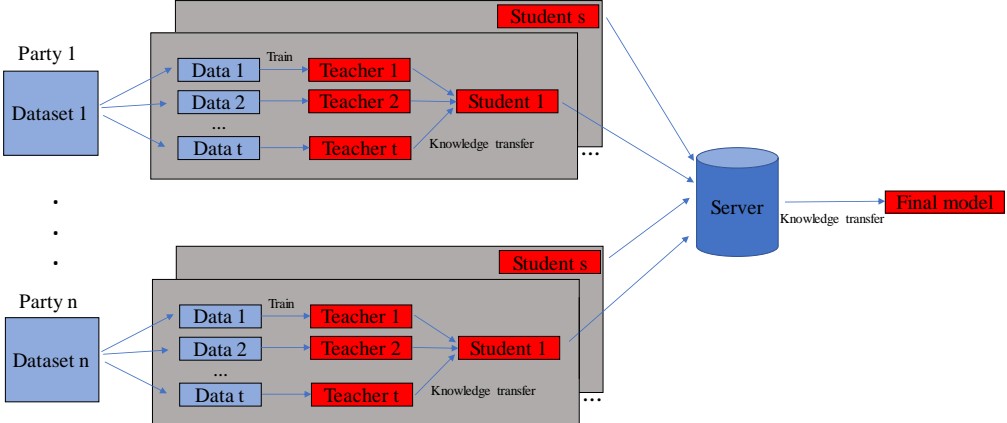

Figure 1: The framework of FedKT

# 3 FEDERATED LEARNING VIA KNOWLEDGE TRANSFER

**Problem Statement** Suppose there are $n$ parties $P_1, .., P_n$. We use $\mathcal{D}^i$ to denote the dataset of $P_i$. With the help of a central server and a public unlabelled dataset $\mathcal{D}_{aux}$, our objective is to build a machine learning model over the datasets $\bigcup_{i \in [n]} \mathcal{D}^i$ without exchanging the raw data. Moreover, the learning process should be able to support three different privacy level settings: (1) $L0$: there is no privacy requirement on the machine learning models. (2) $L1$ (server-noise): in the case where the final model has to be sent back to the parties or even published, it should satisfy differential privacy guarantees to protect against potential inference attacks (Shokri et al., 2017; Fredrikson et al., 2015; Nasr et al., 2019). (3) $L2$ (party-noise): in the case where the server is curious and all the models transferred between the parties and the server during training should satisfy differential privacy guarantees.

**The Overall Framework** The framework of FedKT is shown in Figure 1. Overall, FedKT adopts a 2-tier PATE structure. On the party side, each party uses knowledge transfer to learn student models and sends them to the server. On the server side, the server takes the received models as teachers to learn a final model using knowledge transfer again. The final model is sent back to the parties and used for predictions. Next, we introduce these two steps in details.

**Learning Student Models on the Parties** Locally, each party has to create $s$ ($s \geq 1$) partitions and learn a student model on each partition. Each partition handles the entire local dataset. Since the operations in each partition are similar, here we describe the process in one partition for ease of presentation. Inside a partition, the local dataset is divided into $t$ disjoint subsets. We train a teacher model separately on each subset, denoted as $T_1, ..., T_t$. Then, the ensemble of teacher models is used to make predictions on the public dataset $\mathcal{D}_{aux}$. For an example $\mathbf{x} \in \mathcal{D}_{aux}$, the *vote count* of class $m$ is the number of teachers that predicts $m$, i.e., $v_m(\mathbf{x}) = |\{i : i \in [t], T_i(\mathbf{x}) = m\}|$. The prediction result of the ensemble is the class that has the maximum vote counts, i.e., $f(\mathbf{x}) = \arg\max_m v_m(\mathbf{x})$. Then, we use the public dataset $\mathcal{D}_{aux}$ with the predicted labels to train a student model. For each partition, we get a student model with the above steps. After all the student models are trained, the parties send their student models to the server for further processing.

**Learning the Final Model on the Server** Suppose the student models of party $i$ are denoted as $U_1^i, ..., U_s^i$ ($i \in [n]$). After receiving all the student models, like the steps on the party side, the server can use these student models as an ensemble to make predictions on the public dataset $\mathcal{D}_{aux}$. The public dataset with the predicted labels is used to train the final model. Here we introduce a technique named *consistent voting* for computing the vote counts of each class. If the student models of a party make the same prediction on an example, we take their predictions into account. Otherwise, the party is not confident at predicting this example and thus we ignore the predictions of its student models. Formally, given an example $\mathbf{x} \in \mathcal{D}_{aux}$, we first compute the vote count of class $m$ on the student

models of party $i$ as $v_m^i(\mathbf{x}) = |\{k : k \in [s], U_k^i(\mathbf{x}) = m\}|$. Next, with consistent voting, the final vote count of class $m$ on all parties is computed as $v_m(\mathbf{x}) = s \cdot |\{i : i \in [n], v_m^i(\mathbf{x}) = s\}|$.

**Differentially Private Versions of FedKT** FedKT can easily satisfy differential privacy guarantees by providing differentially private prediction results to the query dataset. Given the privacy parameter $\gamma$, we can add noises to the vote count histogram such that $f(\mathbf{x}) = \arg\max_m\{v_m(\mathbf{x}) + Lap(\frac{1}{\gamma})\}$, where $Lap(\frac{1}{\gamma})$ is the noises generated from Laplace distribution with location 0 and scale $\frac{1}{\gamma}$. Note that we do not need to add noises on both the parties and the server. For the $L1$ setting, we only need to add noises on the server side. The parties can train and send non-differentially private student models to the server. For the $L2$ setting, we only need to add noises on the party side so that the student models are differentially private. Then, the final model naturally satisfies differential privacy guarantees. More analysis on privacy loss will be presented in Section 4.

Algorithm 1 shows the whole training process of FedKT. In the algorithm, for each party (i.e., Line 1) and its each partition (i.e., Line 3), we train a student model using knowledge transfer (i.e., Lines 4-12). The student models are sent to the server. Then, the server trains the final model using knowledge transfer again (i.e., Lines 14-23). For different privacy level settings, we have the corresponding noises injection operations on the server side (i.e., Lines 20-21) or the party side (i.e., Lines 9-10).

**Overhead Analysis of FedKT** Suppose the size of each model is $M$. Then, the total communication size of FedKT is $nsM + nM = nM(s+1)$ including sending the student models to the server and sending back the final model to the parties. Suppose the number of communication rounds in FedAvg is $r$ and all the parties participate in the training in every iteration. Then the total communication size of FedAvg is $2nMr$. Thus, when $r > \frac{s+1}{2}$, the communication cost of FedAvg is higher than FedKT. This value can be quite small, e.g., $r = 2$ if we set $s = 2$. As we will show in the experiments, FedKT can achieve much better performance than FedAvg give the same communication size constraint. The computation overhead of FedKT is usually larger than FedAvg since FedKT needs to train many teacher and student models. It also requires the parties to be able to store multiple models in the training process. Thus, FedKT is more suitable for the cross-silo setting (Yang et al., 2019), where the parties (e.g., companies, data centers) have a relatively large computation power and storage capacity.

**Applications** In our method, we assume that we have access to a public unlabelled dataset. Note that the assumption will not greatly restrict our method's applicability. The public datasets widely exist especially for texts and images including medical data (e.g., Kaggle COVID-19 Open Research Dataset Challenge, Chest X-Ray Images). Thus, our approach can be applied in typical computer vision tasks, natural language processing tasks, and even healthcare tasks. Nevertheless, if the data are very sensitive and we cannot access to any public data, it is still possible to generate synthetic data using GAN with privacy guarantees (Yoon et al., 2019; Torkzadehmahani et al., 2019) on the parties. The synthetic data can be used as public data in our approach. Moreover, we have conducted experiments on a real-world scenario. We use one chest X-ray dataset as the private dataset and another chest X-ray dataset as a public dataset to simulate a real-world federated case. The experiments are available at Appendix B.7, which demonstrates that FedKT does not have strict limitation on the distribution of public data.

Besides the typical applications, the feasibility of single-round federated learning can enable new applications, e.g., model as a service. One can sell or buy the student models in a model market. Then, after the model trade, a final model can be learned using the student models without further communication.

**Integration with Other Federated Learning Algorithms** While FedKT is designed with a round-optimal goal, it is still applicable even if there is no limitation of communication rounds. We can use FedKT to train a model as an initialization step. The model can be seen as a initialized global model, and then we can continue to conduct iterative federated learning using the standard approaches (e.g., FedAvg, FedProx) based on it. As shown in Section 5.2, the communication rounds can be significantly reduced to achieve the same accuracy by applying FedKT as model initialization.

---

**Algorithm 1:** The FedKT algorithm

---

**Input:** local datasets $\mathcal{D}^1, ..., \mathcal{D}^n$ of parties $P_1, ..., P_n$, number of partitions $s$ in each party, number of subsets $t$ in each partition, number of classes $u$, public query dataset $\mathcal{D}_{aux}$, privacy parameter $\gamma$, privacy level $l$

**Output:** The final model $F$.

1 **for** $i = 1, ..., n$ **do**
    `/* Conduct on party` $P_i$ `*/`
2     Create $s$ partitions (i.e., $D_1^i, ..., D_s^i$) on dataset $\mathcal{D}^i$ such that $D_j^i = \bigcup_{k \in [t]} D_{j,k}^i$ for all $j \in [s]$, where $D_{j,k}^i$ is a subset.
3     **for** $j = 1, ..., s$ **do**
4         **for** $k = 1, ..., t$ **do**
5             Train a teacher model $T_{j,k}^i$ on subset $D_{j,k}^i$.
6         **for** $all$ $\mathbf{x} \in \mathcal{D}_{aux}$ **do**
7             **for** $all$ $m \in [u]$ **do**
8                 $v_m(\mathbf{x}) \leftarrow |\{k : k \in [t], T_{j,k}^i(\mathbf{x}) = m\}|$
9                 **if** $l == L2$ **then**
10                     $v_m(\mathbf{x}) \leftarrow v_m(\mathbf{x}) + Lap(1/\gamma)$
11             $f(\mathbf{x}) = \arg\max_m v_m(\mathbf{x})$
12         Train a student model $U_j^i$ on dataset $\{(\mathbf{x}, f(\mathbf{x}))\}_{\mathbf{x} \in \mathcal{D}_{aux}}$.
13     Send the student models $\{U_j^i : j \in [s]\}$ to the server.
    `/* Conduct on the server */`
14 **for** $all$ $\mathbf{x} \in \mathcal{D}_{aux}$ **do**
15     **for** $all$ $i \in [n]$ **do**
16         **for** $all$ $m \in [u]$ **do**
17             $v_m^i(\mathbf{x}) \leftarrow |\{k : k \in [s], U_k^i(\mathbf{x}) = m\}|$
18     **for** $all$ $m \in [u]$ **do**
19         $v_m(\mathbf{x}) \leftarrow s \cdot |\{i : i \in [n], v_m^i(\mathbf{x}) = s\}|$
20         **if** $l == L1$ **then**
21             $v_m(\mathbf{x}) \leftarrow v_m(\mathbf{x}) + Lap(1/\gamma)$
22     $f(\mathbf{x}) = \arg\max_m v_m(\mathbf{x})$
23 Train the final model $F$ on dataset $\{(\mathbf{x}, f(\mathbf{x}))\}_{\mathbf{x} \in \mathcal{D}_{aux}}$.

---

## 4 DATA-DEPENDENT PRIVACY ANALYSIS OF FEDKT

In this section, we use the moments accountant method (Abadi et al., 2016) to track the privacy loss in the training process. In addition to the example-level differential privacy, we also consider the party-level differential privacy (Geyer et al., 2017; McMahan et al., 2018), which is more strict and attractive in the federated setting. Instead of aiming to protect a single record, party-level differential privacy ensures that the learned model does not reveal whether a party participated in federated learning or not. Next, we mainly present the party-level differential privacy analysis for FedKT-L1 and example-level differential privacy analysis for FedKT-L2. For more analysis and proofs of the theorems in this section, please refer to Appendix A.

**FedKT-L1** Considering we change the whole dataset of a party, then at most $s$ student models will be influenced. Thus, on the server side, the sensitivity of the vote count histogram is $2s$ (i.e., the vote count of a class increases by $s$ and the vote count of another class decreases by $s$). According to the Laplace mechanism, we have the following theorem.

**Theorem 1.** Let $\mathcal{M}$ be the $f$ function executed on the server side. Given the number of partitions $s$ and the privacy parameter $\gamma$, $\mathcal{M}$ satisfies $(2s\gamma, 0)$ party-level differential privacy.

Given multiple queries, an straightforward way to compute the privacy loss is to use the advanced composition (Dwork et al., 2014). According to Papernot et al. (2017), we can get a tighter bound of the privacy loss by conducting a data-dependent privacy analysis with the moments accountant method. Similarly, we conduct a data-dependent privacy analysis for FedKT.

**Theorem 2.** Let $\mathcal{M}$ be $(2s\gamma, 0)$ party-level differentially private. Let $q \geq \Pr[\mathcal{M}(d) \neq o^*]$ for some outcome $o^*$. Let $l, \gamma \geq 0$ and $q < \frac{e^{2s\gamma}-1}{e^{4s\gamma}-1}$. For any aux and any two party-adjacent datasets $d$ and $d'$, $\mathcal{M}$ satisfies

$$\alpha_{\mathcal{M}}(l; \mathsf{aux}, d, d') \leq \min(\log((1-q)\Big(\frac{1-q}{1-e^{2s\gamma}q}\Big)^l + qe^{2s\gamma l}), 2s^2\gamma^2 l(l+1)).$$

Here $\Pr[\mathcal{M}(d) \neq o^*]$ can be bounded by Lemma 4 of Papernot et al. (2017).

With Theorem 2, we can track the privacy loss of each query (Abadi et al., 2016).

**FedKT-L2** For each partition on the party side, we add Laplace noises to the vote counts, which is same with the PATE approach. Thus, we have the following theorem.

**Theorem 3.** Let $\mathcal{M}$ be the $f$ function executed on each partition of a party. Let $q \geq \Pr[\mathcal{M}(d) \neq o^*]$ for some outcome $o^*$. Let $l, \gamma \geq 0$ and $q < \frac{e^{2\gamma}-1}{e^{4\gamma}-1}$. For any aux and any two adjacent datasets $d$ and $d'$, $\mathcal{M}$ satisfies

$$\alpha_{\mathcal{M}}(l; \mathsf{aux}, d, d') \leq \min(\log((1-q)\Big(\frac{1-q}{1-e^{2\gamma}q}\Big)^l + qe^{2\gamma l}), 2\gamma^2 l(l+1)).$$

After bounding the privacy loss of each party, we can use the parallel composition to bound the privacy loss of the final model.

**Theorem 4.** Suppose the student models of party $P_i$ satisfy $(\varepsilon_i, \delta)$-differential privacy. Then, the final model satisfies $(\max_i \varepsilon_i, \delta)$-differential privacy.

Note that the above privacy analysis is data-dependent. Thus, the final privacy budget is also data-dependent and may have potential privacy breaches if we publish the budget. Like previous studies (Papernot et al., 2017; Jordon et al., 2019), we report the data-dependent privacy budgets in the experiments. As future work, we plan to use the smooth sensitivity algorithm (Nissim et al., 2007) to add noises to the privacy losses. Also, we may get a tighter bound of the privacy loss if adopting the Gaussian noises (Papernot et al., 2018).

## 5 EXPERIMENTS

To evaluate FedKT, we conduct experiments on four public datasets: (1) A random forest on *Adult* dataset. The number of trees is set to 100 and the maximum tree depth is set to 6. (2) A gradient boosting decision tree (GBDT) model on *cod-rna* dataset. The maximum tree depth is set to 6. (3) A CNN on *MNIST* dataset. The CNN has two 5x5 convolution layers followed with 2x2 max pooling (the first with 6 channels and the second with 16 channels), two fully connected layers with ReLu activation (the first with 120 units and the second with 84 units), and a final softmax output layer. (4) The same CNN on extended *SVHN* dataset. For the first two datasets, we split the original dataset at random into train/test/public sets with a 75%/12.5%/12.5% proportion. For MNIST and SVHN, we use one half of the original test dataset as the public dataset and the remaining as the final test dataset. Experiments on two additional datasets, CelebA and chest X-ray images, are available at Appendix B.7. Like previous studies (Yurochkin et al., 2019; Wang et al., 2020), we use the Dirichlet distribution to simulate the heterogeneous data partition among the parties. Suppose there are $n$ parties. We sample $p_k \sim Dir_n(\beta)$ and allocate a $p_{k,j}$ proportion of the instances of class $k$ to party $j$, where $Dir(\beta)$ is the Dirichlet distribution with a concentration parameter $\beta$ (0.5 by default). By default, we set the number of parties to 50 for Adult and cod-rna and to 10 for MNIST and SVHN. We set $s$ to 2 and $t$ to 5 by default for all datasets. For more details in the experimental settings and guidelines to set the hyper-parameters (i.e., $s$ and $t$), please refer to Appendix B.

Table 1: The test accuracy comparison between FedKT and the other baselines.

| Datasets | FedKT | SOLO | PATE | XGBoost | FedAvg | | FedProx | | SCAFFOLD | |
|---|---|---|---|---|---|---|---|---|---|---|
| | | | | | 2 rounds | 50 rounds | 2 rounds | 50 rounds | 2 rounds | 50 rounds |
| Adult | 82.2% | 68.6% | 83.5% | ↘ | ↘ | ↘ | ↘ | ↘ | ↘ | ↘ |
| cod-rna | 88.3% | 65.0% | 91.1% | 91.2% | ↘ | ↘ | ↘ | ↘ | ↘ | ↘ |
| MNIST | 95.9% | 80.0% | 97.8% | ↘ | 83.5% | 98.0% | 83.0% | 98.6% | 74.2% | 98.4% |
| SVHN | 83.2% | 62.8% | 86.6% | ↘ | 58.4% | 80.3% | 59.9% | 86.9% | 15.6% | 84.5% |

We compare FedKT with six baselines: (1) SOLO: each party trains its model locally and does not participate in federated learning. (2) PATE (Papernot et al., 2017): we use the PATE framework to train a final model on all data in a centralized setting (i.e., only a single party with the whole dataset) without adding noises. This method defines an upper bound of learning a final model using knowledge transfer with public unlabelled data. (3) XGBoost (Chen & Guestrin, 2016): the XGBoost algorithm for the GBDT model on the whole dataset in a centralized setting. This method defines an upper bound of learning the GBDT model. (4) FedAvg (McMahan et al., 2016); (5) FedProx (Li et al., 2020c); (6) SCAFFOLD (Karimireddy et al., 2020). Here (4)-(6) are three standard or state-or-the-art federated learning algorithms.

## 5.1 EFFECTIVENESS

Table 1 shows the accuracy of FedKT[1] compared with the other baselines. For SOLO, we report the average accuracy of the parties. For FedAvg, FedProx, and SCAFFOLD, we report the results of running for 2 rounds (i.e., when the baselines start to have a larger communication cost than FedKT as shown in previous overhead analysis) and 50 rounds. From this table, we have the following observations. First, except for FedKT, the other federated learning algorithms can only learn specific models. FedKT is able to learn all the studied models with a significant improvement compared with SOLO. Second, the accuracy of FedKT is close to PATE, which means our design has little accuracy loss compared with the centralized learning setting. Third, given the limited communication size constraint, FedKT can achieve much better accuracy than FedAvg, FedProx, and SCAFFOLD. The performance of the iterative methods is poor running with two rounds. Last, FedKT is just slightly worse than FedAvg, FedProx, and SCAFFOLD running for 50 rounds. FedKT gives promising results even with a single round communication.

## 5.2 COMMUNICATION EFFICIENCY

Figure 2 shows the test accuracy with varying the number of communication rounds/communication size for MNIST and SVHN. We do not present the results for Adult and cod-rna since there is no baseline running in a federated setting. For MNIST, FedAvg, FedProx, and SCAFFOLD need about 23 rounds, 7 rounds, and 13 rounds to achieve a better accuracy than FedKT, respectively. For SVHN, FedAvg, FedProx, and SCAFFOLD need about 17, 18, and 33 rounds to achieve a better accuracy than FedKT, respectively. Moreover, the communication size of FedKT is close to 0 (see the black points in Figure 2c and Figure 2d). FedKT only needs 5.4M and 7.5M communication cost on MNIST and SVHN, respectively. Overall, FedKT is much more communication-efficient than the other approaches.

In Figure 2, FedKT-Prox is an approach that applies FedKT to learn a initialized global model and then applies FedProx to conduct iterative learning. As we can see, FedKT-Prox is always better than the other federated learning approaches. The communication round can be significantly reduced to achieve the same accuracy if adopting FedKT as initialization. For example, for SVHN, FedKT-Prox only needs about 16 rounds to achieve 87% accuracy, while FedProx needs about 40 rounds.

## 5.3 DIFFERENTIAL PRIVATE FEDKT

We run FedKT with different $\gamma$ and number of queries. When running FedKT-L1, we tune the percentage of number of queries on the server side. When running FedKT-L2, on the contrary, we tune the percentage of number of queries on the party side. The selected results on Adult and cod-rna are reported in Table 2. While differentially private FedKT does not need any knowledge on

---

[1]For simplicity, we use FedKT to denote FedKT-L0, unless specified otherwise.

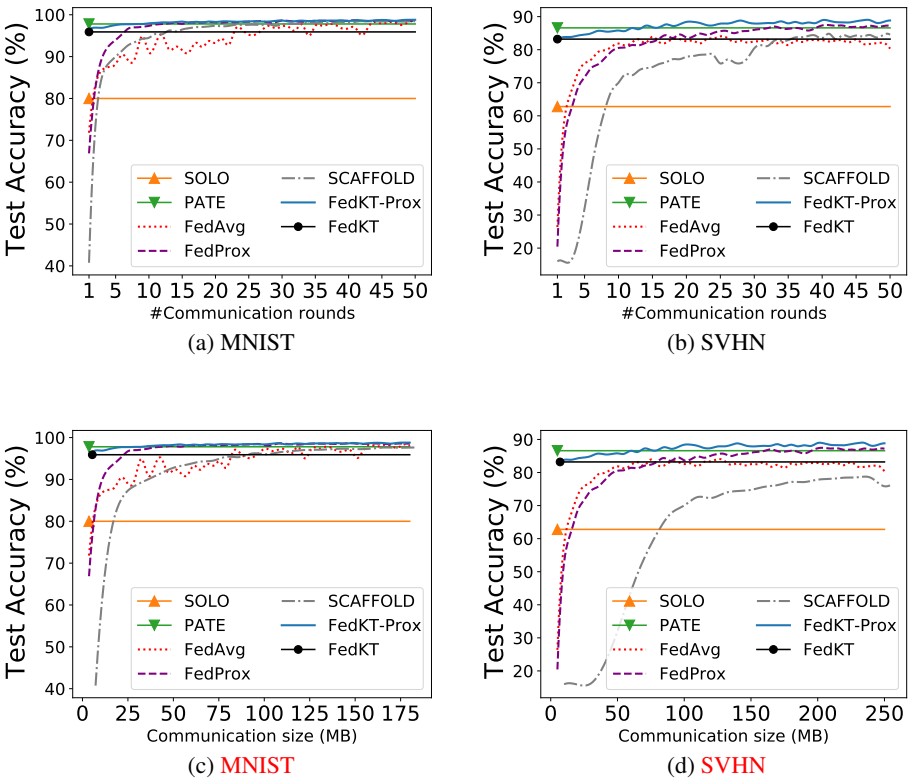

Figure 2: The test accuracy with increasing communication rounds/communication size.

Table 2: The privacy loss and test accuracy of FedKT-L1 and FedKT-L2 given different $\gamma$ and number of queries. L0 acc is the test accuracy of FedKT-L0.

| Datasets | FedKT-L1 | | | | L0 acc | FedKT-L2 | | | | L0 acc |
|---|---|---|---|---|---|---|---|---|---|---|
| | $\gamma$ | #queries | $\varepsilon$ | acc | | $\gamma$ | #queries | $\varepsilon$ | acc | |
| Adult | 0.04 | 0.5% | 2.56 | 76.8% | 82.2% | 0.05 | 0.5% | 3.24 | 79.0% | 82.4% |
| | 0.04 | 1.0% | 4.73 | 80.2% | | 0.05 | 1.0% | 4.76 | 79.2% | |
| cod-rna | 0.06 | 0.5% | 5.48 | 82.6% | 88.3% | 0.05 | 0.5% | 4.51 | 81.4% | 89.7% |
| | 0.1 | 0.5% | 6.89 | 84.7% | | 0.05 | 2.0% | 9.78 | 84.7% | |

the model architecture, the accuracy is still comparable to the non-private version given a privacy budget less than 10. For more results, please refer to Appendix B.8.

## 6 CONCLUSIONS

In this paper, we propose FedKT, a model-agnostic federated learning algorithm with a single communication round. Our experiments show that FedKT can learn different models with a comparable or even better accuracy compared with the other state-of-the-art algorithms. Moreover, the accuracy of differentially private FedKT is comparable to the non-differentially private version with a modest privacy budget. Instead of updating and aggregating in an iterative way, our work show that it is promising to adopt knowledge transfer to achieve round-optimal communication efficiency. For future work, we will extend FedKT to the cross-device setting, where the number of parties is large and the data size of each party is small.

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

APPENDIX

In this appendix, we first present the data-dependent privacy analysis of FedKT-L1 and FedKT-L2 in Section A. Next, in Section B.1, we show the details of our experimental settings. Then, we show additional experimental results in Section B.2 to Section B.8. Specifically, in Section B.2, we study the hyper-parameters setting of FedKT. In Section B.3, we study the performance of FedKT with different levels of imbalance. In Section B.4, we study the performance of FedKT with different number of parties. In Section B.6, we study the effect of our consistent voting technique. In Section B.7, we conduct experiments on addtional datasets, including the CelebA dataset with VGG-9 and two chest X-Ray datasets with CNN. Last, in section B.8, we show the performance of FedKT-L1 and FedKT-L2.

## A    PRIVACY ANALYSIS OF FEDKT

In this section, we analyze the privacy loss of FedKT-L1 and FedKT-L2 using the moments accountant method (Abadi et al., 2016).

### A.1    PRELIMINARIES

We first introduce two theorems from the previous studies, which will be used in our analysis. The first theorem is from Bun & Steinke (2016) and the second theorem is from Papernot et al. (2017).

**Theorem 5.** Let $\mathcal{M}$ be $(2\gamma, 0)$-differentially private. For any $l$, aux, neighboring inputs $d$ and $d'$, we have

$$\alpha_{\mathcal{M}}(l; \mathsf{aux}, d, d') \leq 2\gamma^2 l(l+1)$$

**Theorem 6.** Let $\mathcal{M}$ be $(2\gamma, 0)$-differentially private and $q \geq \Pr[\mathcal{M}(d) \neq o^*]$ for some outcome $o^*$. Let $l, \gamma \geq 0$ and $q < \frac{e^{2\gamma}-1}{e^{4\gamma}-1}$. Then for any aux and any neighbor $d'$ of $d$, $\mathcal{M}$ satisfies

$$\alpha_{\mathcal{M}}(l; \mathsf{aux}, d, d') \leq \log((1-q)\Big(\frac{1-q}{1-e^{2\gamma}q}\Big)^l + qe^{2\gamma l})$$

$\Pr[\mathcal{M}(d) \neq o^*]$ can be bounded by the following lemma.

**Lemma 7.** Let $\mathbf{v}$ be the label score vector for an instance $d$ with $v_{o^*} \geq v_o$ for all $o$. Then

$$\Pr[\mathcal{M}(d) \neq o^*] \leq \sum_{o \neq o^*} \frac{2 + \gamma(v_{o^*} - v_o)}{4 \exp(\gamma(v_{o^*} - v_o))}$$

The following theorem from Abadi et al. (2016) can be used to convert the moments into $(\varepsilon, \delta)$-differential privacy.

**Theorem 8.** 1. [Composability] Suppose that a mechanism $\mathcal{M}$ consists of a sequence of adaptive mechanisms $\mathcal{M}_1, \ldots, \mathcal{M}_k$ where $\mathcal{M}_i \colon \prod_{j=1}^{i-1} \mathbb{R}_j \times \mathcal{D} \to \mathbb{R}_i$. Then, for any $\lambda$

$$\alpha_{\mathcal{M}}(\lambda) \leq \sum_{i=1}^{k} \alpha_{\mathcal{M}_i}(\lambda),$$

2. [Tail bound] For any $\varepsilon > 0$, the mechanism $\mathcal{M}$ is $(\varepsilon, \delta)$-differentially private for

$$\delta = \min_{\lambda} \exp(\alpha_{\mathcal{M}}(\lambda) - \lambda\varepsilon).$$

### A.2    PROOF OF THEOREM 2

*Proof.* According to Theorem 1, we can get Theorem 2 by replacing $\gamma$ in Theorem 5 and Theorem 6 to $2s\gamma$. □

### A.3 Proof of Theorem 3

*Proof.* The noises injection on the party side is similar to the PATE approach. The sensitivity of the $f$ function executed on the party side is 2. Thus, we can get the theorem by combining Theorem 5 and Theorem 6. □

### A.4 Example-Level Differential Privacy Analysis of FedKT-L1

Here we analyze the example-level differential privacy of FedKT-L1. If we change a single example of the original dataset (i.e., the union of all the local datasets), only a single party will be influenced. More precisely, for each partition of the party, only a single teacher model will be influenced. Then, even though changing a single record, the student model is still unchanged if the top-2 vote counts of the teachers differ at least 2. Thus, if not applying consistent voting, we have the following theorem.

**Theorem 9.** Let $\mathcal{M}$ be the $f$ function executed in the server. Let $q \geq \Pr[\mathcal{M}(d) \neq o^*]$ for some outcome $o^*$. Let $\mathcal{D}_{aux}$ denotes the query dataset. Given a query $i$, suppose the top-2 vote counts are $v_1^i$ and $v_2^i$. In party $P_i$, let $z_i$ denotes the number of partitions that there $\exists q \in \mathcal{D}_{aux}$ such that $v_1^q - v_2^q \leq 1$ when training the student model. Let $z = max_i z_i$. Let $l, \gamma \geq 0$ and $q < \frac{e^{2z\gamma} - 1}{e^{4z\gamma} - 1}$. We have

$$\alpha_{\mathcal{M}}(l; \mathsf{aux}, d, d') \leq \min(\log((1-q)\Big(\frac{1-q}{1-e^{2z\gamma}q}\Big)^l + qe^{2z\gamma l}), 2z^2\gamma^2 l(l+1)) \tag{4}$$

*Proof.* Given a query dataset $\mathcal{D}_{aux}$, $z$ is the number of partitions such that there exists a query that the top-2 vote counts differ at most 1. In other words, there are at most $z$ student models will be changed if we change a single record of the original dataset. Thus, the vote counts change by at most $2z$ on the server side and $\mathcal{M}$ is $(2z\gamma, 0)$-differentially private with respect to $d$ and $\mathcal{D}_{aux}$. Then, we can get this theorem by replacing $\gamma$ of Theorem 5 and Theorem 6 to $z\gamma$. □

Note that the example-level differential privacy is same as party-level differential privacy when $z = s$. Also, if we applied consistent voting in FedKT-L1, the vote counts may change by $s$ even if only a single student model is affected. Thus, the example-level differential privacy of FedKT-L1 is usually same as party-level differential privacy.

### A.5 Party-Level Differential Privacy Analysis of FedKT-L2

Here we study party-level differential privacy of FedKT-L2.

**Theorem 10.** Let $\mathcal{M}$ be the $f$ function executed on each partition of a party. Given the number of subsets in each partition $t$ and the privacy parameter $\gamma$, $\mathcal{M}$ satisfies $(2t\gamma, 0)$ party-level differential privacy.

*Proof.* Considering changing the whole local dataset, then $t$ teachers will be influenced and the vote counts change by at most $2t$. Thus, from a party-level perspective, the sensitivity of the vote counts is $2t$ and $\mathcal{M}$ satisfies $(2t\gamma, 0)$-differential privacy. □

Like Theorem 9, we have the following theorem to track the moments.

**Theorem 11.** Let $\mathcal{M}$ be the $f$ function executed on the party side and $q \geq \Pr[\mathcal{M}(d) \neq o^*]$ for some outcome $o^*$. Suppose the number of subsets in a partition is $t$. Let $l, \gamma \geq 0$ and $q < \frac{e^{2t\gamma} - 1}{e^{4t\gamma} - 1}$. Then for any $\mathsf{aux}$ and any two party-adjacent datasets $d$ and $d'$, $\mathcal{M}$ satisfies

$$\alpha_{\mathcal{M}}(l; \mathsf{aux}, d, d') \leq \min(\log((1-q)\Big(\frac{1-q}{1-e^{2t\gamma}q}\Big)^l + qe^{2t\gamma l}), 2t^2\gamma^2 l(l+1)).$$

Note that the party-level privacy loss of FedKT-L2 can be quite large. To mitigate the impact of the noises, we usually expect $t$ to be large to have a tighter bound of $\Pr[\mathcal{M}(d) \neq o^*]$. However, when $t$ is large, the bound in Theorem 11 is also large. In fact, since every student model satisfies differential privacy, it is not necessary to apply party-level differential privacy in FedKT-L2.

Table 3: The datasets used and their learning models. The detailed model structures are shown in the paper.

| Datasets | Adult | cod-rna | MNIST | SVHN |
|---|---|---|---|---|
| #Training examples | 24421 | 54231 | 50000 | 604288 |
| #Public examples | 4070 | 9039 | 5000 | 13016 |
| #Test examples | 4070 | 9039 | 5000 | 13016 |
| #Classes | 2 | 2 | 10 | 10 |
| #Parties (by default) | 50 | 50 | 10 | 10 |
| Model | Random Forest | GBDT | CNN | CNN |
| Implemented library | scikit-learn 0.22.1 | XGBoost 1.0.2 | PyTorch 1.6.0 | PyTorch 1.6.0 |

Table 4: The default parameters used in our experiments.

| | Parameters | Adult | cod-rna | MNIST | SVHN |
|---|---|---|---|---|---|
| common | #parties | 50 | 50 | 10 | 10 |
| | tree depth | 6 | 6 | ╲ | ╲ |
| | learning rate | ╲ | 0.05 | 0.001 | 0.001 |
| | batch size | ╲ | ╲ | 32 | 64 |
| | #epochs | ╲ | ╲ | 10 | 10 |
| FedKT | number of partitions in a party | 2 | 2 | 2 | 2 |
| | number of subsets in a partition | 5 | 5 | 5 | 5 |
| FedProx | regularization term $\mu$ | ╲ | ╲ | 0.1 | 0.1 |

# B   EXPERIMENTS

## B.1   ADDITIONAL DETAILS OF EXPERIMENTAL SETTINGS

We use *Adult*, *cod-rna*, *MNIST*, and *SVHN* for our experiments, where *Adult* and *cod-rna* are downloaded from this link[2]. The details of the datasets are shown in Table 3. Besides these four datasets, in section B.7, we also conduct experiments on CelebA with VGG-9, and chest X-ray datasets with CNN.

We compare FedKT with the other six baselines. For FedAvg, FedProx, and SCAFFOLD, all parties participate in federated learning for each round (i.e., no party sampling). The learning rate is tuned from $\{0.001, 0.01\}$ and the number of local epochs is tuned from $\{10, 20, 40\}$. We found that the baselines can get best performance when the learning rate is set to 0.001 and the number of local epochs is set to 10. For FedProx, the regularization term $\mu$ is tuned from $\{0.1, 1\}$. For SCAFFOLD, same as the experiments of the paper (Karimireddy et al., 2020), we use option II to calculate the control variates. For FedKT, SOLO, and PATE, the number of local epochs is simply set to 100. For PATE, the number of teacher models is set to be same as the number of parties of FedKT. We use the Adam optimizer and the $L_2$ regularization is set to $10^{-6}$. The final parameters are summarized in Table 4.

## B.2   HYPER-PARAMETERS STUDY

### B.2.1   NUMBER OF PARTITIONS IN EACH PARTY

Here we study the impact of the number of partitions (i.e., the parameter $s$) on FedKT. Table 5 shows the test accuracy of FedKT with different $s$. From Table 5, we can see that the accuracy can be improved if we increase $s$ from 1 to 2. However, if we further increase $s$, there is little or no improvement on the accuracy while the communication and computation overhead is larger. Thus, from our empirical study, we suggest users to simply set $s$ to 2 for FedKT-L0 if they do not want to tune the parameters. For FedKT-L1 and FedKT-L2, since the privacy loss increases as $s$ increases, we suggest to set $s$ to small values. Users can simply set $s$ to 1 or tune $s$ from small values (i.e., 1 or 2) to find a best accuracy-privacy trade-off.

---

[2] https://www.csie.ntu.edu.tw/~cjlin/libsvmtools/datasets/

Table 5: The test accuracy of FedKT with number of partitions ranging between 1 and 5. We run 5 trials and report the mean and standard deviation. The number of subsets in each partition is set to 5 by default.

| #partitions | 1 | 2 | 3 | 4 | 5 |
|---|---|---|---|---|---|
| Adult | $80.8\% \pm 1.4\%$ | **82.0%** $\pm 0.6\%$ | $81.5\% \pm 0.6\%$ | $81.2\% \pm 0.5\%$ | $81.1\% \pm 0.1\%$ |
| cod-rna | $87.7\% \pm 0.6\%$ | **88.3%** $\pm 0.6\%$ | **88.3%** $\pm 0.5\%$ | $88.1\% \pm 0.5\%$ | $88.2\% \pm 0.5\%$ |
| MNIST | $94.5\% \pm 0.5\%$ | **96.0%** $\pm 0.4\%$ | $95.5\% \pm 0.4\%$ | $95.8\% \pm 0.3\%$ | $95.5\% \pm 0.3\%$ |
| SVHN | $81.5\% \pm 0.6\%$ | $83.2\% \pm 0.4\%$ | **83.5%** $\pm 0.4\%$ | $83.5\% \pm 0.4\%$ | $83.4\% \pm 0.3\%$ |

Table 6: The test accuracy of FedKT with number of subsets in each partition ranging between 5 and 20. We run 5 trials and report the mean and standard deviation. For Adult, since there is a party with less than 15 examples, the experiment cannot successfully run when the number of subsets is not smaller than than 15. For cod-rna, since there is a party with less than 20 examples, the experiment cannot successfully run when setting the number of subsets to 20. The number of partitions in each party is set to 2 by default.

| #subsets | 5 | 10 | 15 | 20 |
|---|---|---|---|---|
| Adult | **82.0%** $\pm 0.6\%$ | $81.1\% \pm 0.7\%$ | ╲ | ╲ |
| cod-rna | **88.3%** $\pm 0.6\%$ | $87.4\% \pm 0.6\%$ | $83.1\% \pm 0.6\%$ | ╲ |
| MNIST | **96.0%** $\pm 0.4\%$ | $93.0\% \pm 0.5\%$ | $92.7\% \pm 0.6\%$ | $90.0\% \pm 0.6\%$ |
| SVHN | **83.2%** $\pm 0.4\%$ | $81.3\% \pm 0.5\%$ | $80.0\% \pm 0.5\%$ | $79.1\% \pm 0.6\%$ |

### B.2.2 NUMBER OF TEACHERS IN EACH PARTITION

Here we study the impact of number of teachers in each partition (i.e., the parameter $t$). Table 6 shows the test accuracy of FedKT with different $t$. As we can see, FedKT can always get the best performance if setting $t$ to 5. If $t$ is large, the size of each data subset is small and the teacher models may not be good at predicting the public dataset. From our empirical study, users can simply set $t$ to 5 if they do not want to tune the parameter. However, if the student models need to satisfy differential privacy (i.e., in FedKT-L2), the privacy loss may potentially be smaller if we increase $t$ according to Lemma 7. Users need to tune $t$ to find the best trade-off between the performance and the privacy loss.

### B.3 IMBALANCE LEVEL

Here we study the impact of the concentration parameter $\beta$ of the Dirichlet distribution in our heterogeneous partition. If $\beta$ is smaller, the data among different parties tend to be more unbalanced. We study three different values: 0.1, 0.5, and 10. The results are shown in Table 7. Given different $\beta$ values, the performance of FedKT is always better than FedAvg and FedProx given limited communication size (i.e., 2 rounds). Also, FedKT is quite stable compared with the other baselines. The results show that FedKT has a good performance under the heterogeneous data partition. FedKT is superior than the other approaches especially when the imbalance level of data distribution is high.

### B.4 NUMBER OF PARTIES

We change the number of parties and compare different approaches. The results are shown in Table 8 and Table 9. FedKT is quite stable with different number of parties compared with the other baselines. Moreover, FedKT is superior than FedAvg, FedProx, and SCAFFOLD with limited communication size in all cases. Even though running the baselines for 50 rounds, the accuracy of FedKT is still comparable.

### B.5 SIZE OF THE PUBLIC DATASET

We show the performance of FedKT with different size of public dataset. The results are shown in Table 10. We can observe that FedKT is stable even though reducing the size of the public dataset. The accuracy decreases no more than 1% and 2% using only 20% of the public dataset on

Table 7: The test accuracy of different approaches with $\beta$ ranging from $\{0.1, 0.5, 10\}$. For Adult and cod-rna, we set the number of parties to 20 so that there are enough training instances for each party to train the teacher models in FedKT when $\beta$ is set to 0.1.

| | $\beta$ | | 0.1 | 0.5 | 10 |
|---|---|---|---|---|---|
| Adult | FedKT | | 81.9% | 82.4% | 82.7% |
| | SOLO | | 57.0% | 68.6% | 82.2% |
| | PATE | | | 83.5% | |
| cod-rna | FedKT | | 88.9% | 89.7% | 90.1% |
| | SOLO | | 58.8% | 67.0% | 83.1% |
| | PATE | | | 91.1% | |
| | XGBoost | | | 91.2% | |
| MNIST | FedKT | | 95.8% | 96.0% | 96.8% |
| | SOLO | | 43.4% | 80.0% | 96.1% |
| | PATE | | | 97.8% | |
| | FedAvg | 2 rounds | 58.2% | 83.5% | 96.7% |
| | | 50 rounds | 85.9% | 98.0% | 99.4% |
| | FedProx | 2 rounds | 56.6% | 83.0% | 93.9% |
| | | 50 rounds | 97.2% | 98.6% | 99.2% |
| | SCAFFOLD | 2 rounds | 44.3% | 74.2% | 83.1% |
| | | 50 rounds | 97.6% | 98.4% | 98.6% |

Table 8: The test accuracy of different approaches with number of parties ranging from $\{10, 20, 30, 40, 50\}$ for Adult and cod-rna.

| | #parties | 10 | 20 | 30 | 40 | 50 |
|---|---|---|---|---|---|---|
| Adult | FedKT | 83.2% | 82.4% | 83.6% | 83.3% | 82.2% |
| | SOLO | 77.5% | 68.6% | 64.4% | 68.3% | 67.0% |
| | PATE | 83.5% | 83.5% | 83.1% | 83.1% | 83.5% |
| cod-rna | FedKT | 88.1% | 89.7% | 86.7% | 87.2% | 88.3% |
| | SOLO | 69.7% | 67.0% | 67.9% | 66.6% | 65.0% |
| | PATE | 91.1% | 90.7% | 90.4% | 91.0% | 91.1% |
| | XGBoost | | | 91.2% | | |

cod-rna and MNIST (i.e., 1807 examples on cod-rna and 1000 examples on MNIST), respectively. Moreover, the accuracy is almost unchanged on Adult.

## B.6 EFFECT OF CONSISTENT VOTING

Here we study the effect of our consistent voting technique. The results are shown in Table 11. From the table, we can observe that the accuracy can be improved about 1%-2.3% by applying consistent voting. Thus, consistent voting is a simple and effective technique.

## B.7 EXPERIMENTS ON ADDITIONAL DATASETS

We conduct experiments on two additional datasets: CelebA and chest X-ray images.

We conduct a gender recognition task on the CelebA dataset. Like MNIST, we use one half of the original test dataset as the public dataset and the remaining as the final test dataset. The number of parties is set to 10. The details of CelebA and the model parameters are shown in Table 12. The model structure of VGG-9 is shown in Table 13.

We use two chest X-ray image datasets to simulate federated learning as a real-world scenario. Specifically, we use Kaggle Chest X-Ray Images [3] as the private dataset and partition it into 5 parties. We use RSNA Pneumonia Detection Challenge dataset [4] as the public dataset for our FedKT

---

[3] https://www.kaggle.com/paultimothymooney/chest-xray-pneumonia
[4] https://www.kaggle.com/c/rsna-pneumonia-detection-challenge

Table 9: The test accuracy of different approaches with number of parties ranging from $\{5, 10, 15, 20\}$ for MNIST.

| | #parties | | 5 | 10 | 15 | 20 |
|---|---|---|---|---|---|---|
| MNIST | FedKT | | 95.2% | 96.0% | 95.4% | 95.8% |
| | SOLO | | 88.1% | 80.2% | 82.1% | 80.1% |
| | PATE | | 98.2% | 97.9% | 97.8% | 97.3% |
| | FedAvg | 2 rounds | 93% | 83.5% | 89.3% | 94.4% |
| | | 50 rounds | 98.7% | 98.0% | 98.9% | 99.0% |
| | FedProx | 2 rounds | 76.9% | 83.0% | 86.2% | 92.6% |
| | | 50 rounds | 98.6% | 98.6% | 98.9% | 99.0% |
| | SCAFFOLD | 2 rounds | 73.2% | 74.2% | 62.9% | 64.5% |
| | | 50 rounds | 98.3% | 98.4% | 98.1% | 97.8% |

Table 10: The test accuracy of FedKT with different size of the public dataset. We vary the portion of the public dataset used in the training from 20% to 100%.

| Datasets | Portion of the public dataset used in training | | | | |
|---|---|---|---|---|---|
| | 20% | 40% | 60% | 80% | 100% |
| Adult | 82.1% | 82.1% | 82.1% | 82.3% | 82.2% |
| cod-rna | 87.3% | 87.6% | 87.8% | 87.8% | 88.3% |
| MNIST | 94.0% | 95.1% | 95.0% | 95.7% | 95.9% |

Table 11: The test accuracy of FedKT with/without consistent voting.

| | Adult | cod-rna | MNIST | SVHN |
|---|---|---|---|---|
| with consistent voting | 82.2% | 88.3% | 96.0% | 83.2% |
| without consistent voting | 81.1% | 87.1% | 93.6% | 81.6% |

algorithm. We use a CNN model (same as the model used in MNIST and SVHN) to predict pneumonia. The number of parties is set to 5. The details of X-Ray task are shown in Table 12.

We try the heterogeneous partition and the experiment results are shown in Table 14. For CelebA, FedKT shows a good performance on a relatively large dataset and deep model. FedKT is better than FedProx and SCAFFOLD even with 50 rounds. Also, the accuracy of FedKT is comparable to FedAvg with 50 rounds. For X-Ray, FedKT has a much better performance than the other federated learning approaches. FedAvg, FedProx, and SCAFFOLD are quite unstable.

Figure 3 shows the test accuracy changing the number of communication rounds. We can find that FedProx and SCAFFOLD are quite unstable on these two tasks. For CelebA, FedKT outperforms FedProx and SCAFFOLD, while FedAvg needs about 25 rounds to achieve a better accuracy than FedKT. This experiments demonstrate the applicability of FedKT on more complex models. For X-ray, FedKT outperforms the other approaches. This experiments show that FedKT does not have strict limitation on the distribution of the public dataset. Even though the public dataset is another dataset differ from the training dataset, which is not partitioned from the global dataset differ from previous experiments, FedKT still has a quite good performance.

## B.8 DIFFERENTIAL PRIVACY

Table 15 and Table 16 present the results of FedKT-L1 and FedKT-L2. Besides the heterogeneous partition ($\beta = 0.5$), we also try homogeneous partition (i.e., the dataset is randomly and equally partitioned into the parties). From the tables, we can see that the accuracy of FedKT-L1 and FedKT-L2 are comparable to the non-private version with a modest privacy budget. Moreover, the moments accountant method usually can achieve a tighter privacy loss than the advanced composition (Dwork et al., 2014). For example, if we run cod-rna under homogeneous data partition setting $\gamma = 0.1$ and the fraction of queries to 1%, the advanced composition gives us $\varepsilon \approx 20.2$ and our analysis gives $\varepsilon \approx 11.2$. Note that the techniques in Papernot et al. (2018) can also be applied to FedKT. For example, we may get a smaller privacy loss if adopting Gaussian noises instead of Laplace

Table 12: The details of tasks on CelebA and Chest X-Ray Image datasets. We conduct gender recognition task on CelebA and pneumonia detection task on X-ray image datasets.

| Datasets | CelebA | Two chest X-ray image datasets |
|---|---|---|
| #Training examples | 162770 | 5216 |
| #Public examples | 9981 | 26684 |
| #Test examples | 9981 | 624 |
| #Classes | 2 | 2 |
| #Parties | 10 | 5 |
| Model | VGG-9 | CNN |
| Implemented library | | PyTorch 1.6.0 |
| Batch size | 128 | 32 |
| #Local epochs for FedAvg/FedProx/SCAFFOLD | 10 | 10 |
| #Epochs for FedKT/SOLO/PATE | 100 | 100 |
| Learning rate | 0.001 | 0.001 |

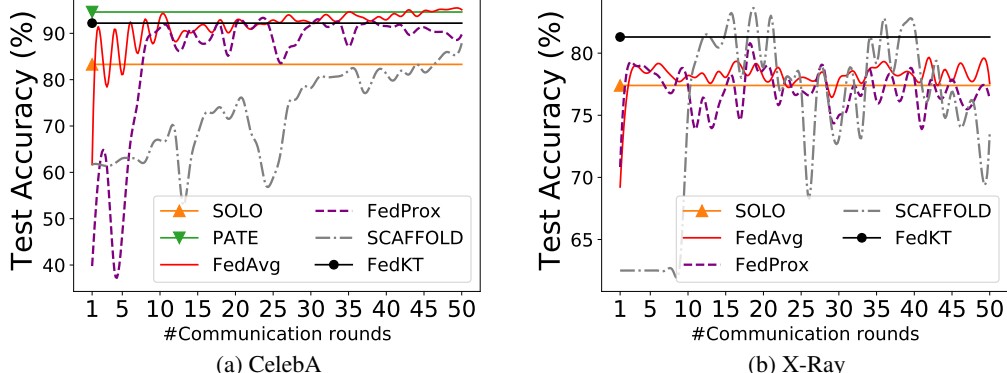

(a) CelebA

(b) X-Ray

Figure 3: The test accuracy with different number of communication rounds. For X-Ray, we do not plot PATE since the line is overlapped with FedKT.

Table 13: The details of VGG-9 architecture. The stride and padding of convolution layers are set to 1 by default. The ReLU activation is applied after each convolution layer and fully connected layer (except the last fully connected layer).

| Layer | Parameters |
|---|---|
| 2D convolution | #input channels: 3, #output channels: 32, kernel size:3 |
| 2D convolution | #input channels: 32, #output channels: 64, kernel size:3 |
| Max pooling | kernel size:2, stride:2 |
| 2D convolution | #input channels: 64, #output channels: 128, kernel size:3 |
| 2D convolution | #input channels: 128, #output channels: 128, kernel size:3 |
| Max pooling | kernel size:2, stride:2 |
| Dropout | probability: 0.1 |
| 2D convolution | #input channels: 128, #output channels: 256, kernel size:3 |
| 2D convolution | #input channels: 256, #output channels: 256, kernel size:3 |
| Max pooling | kernel size:2, stride:2 |
| Dropout | probability: 0.1 |
| Linear | #input features: 4096, #output features: 512 |
| Linear | #input features: 512, #output features: 512 |
| Dropout | probability: 0.1 |
| Linear | #input features: 512, #output features: 2 |

Table 14: The test accuracy of FedKT and the other baselines on CelebA and X-ray datasets. For FedKT, the number of partition in each party is set to 2 and the number of subsets of each partition is set to 10. For FedProx, we set the regularization term $\mu$ to 0.1.

| | FedKT | SOLO | PATE | FedAvg | | FedProx | | SCAFFOLD | |
|---|---|---|---|---|---|---|---|---|---|
| | | | | 2 rounds | 50 rounds | 2 rounds | 50 rounds | 2 rounds | 50 rounds |
| CelebA | 92.2% | 83.3% | 94.6% | 90.5% | 95.1% | 61.4% | 89.7% | 61.7% | 87.8% |
| X-ray | 81.3% | 77.4% | 81.4 % | 77.1% | 77.6% | 78.8% | 76.3% | 62.5% | 73.6% |

noises. Generally, our framework can also benefit from the state-of-the-art approaches on the privacy analysis of PATE, which we may investigate in the future.

Table 15: The accuracy and party-level $\varepsilon$ of FedKT-L1 on Adult and cod-rna given different $\gamma$ values and number of queries. For each setting, we run 3 trials and report the median accuracy and the corresponding $\varepsilon$. The number of partitions is set to 1 and the number of subsets in each partition is set to 5. The failure probability $\delta$ is set to $10^{-5}$.

| | data partitioning | $\gamma$ | #queries | $\varepsilon$ | acc | FedKT-L0 acc |
|---|---|---|---|---|---|---|
| Adult | Heterogeneous | 0.04 | 0.1% | 0.64 | 71.3% | 82.2% |
| | | | 0.5% | 2.56 | 76.8% | |
| | | | 1% | **4.73** | **80.2%** | |
| | | 0.06 | 0.1% | 0.96 | 75.7% | |
| | | | 0.5% | 3.64 | 77.6% | |
| | | | 1% | 5.78 | 80.2% | |
| | | 0.08 | 0.1% | 1.23 | 76.0% | |
| | | | 0.5% | 4.25 | 76.3% | |
| | | | 1% | 7 | 80.3% | |
| | Homogeneous | 0.02 | 0.1% | 0.32 | 72.2% | 82.4% |
| | | | 0.5% | 1.25 | 76.1% | |
| | | | 1% | 1.79 | 80.1% | |
| | | 0.04 | 0.1% | 0.6 | 76.0% | |
| | | | 0.5% | 1.9 | 80.4% | |
| | | | 1% | 3.32 | 81.5% | |
| | | 0.06 | 0.1% | 0.75 | 76.1% | |
| | | | 0.5% | 2.03 | 81.7% | |
| | | | 1% | **3.36** | **82.1%** | |
| cod-rna | Heterogeneous | 0.04 | 0.1% | 1.09 | 66.8% | 88.3% |
| | | | 0.5% | 3.54 | 72.5% | |
| | | | 1% | 5.14 | 75.2% | |
| | | 0.06 | 0.1% | 1.52 | 69% | |
| | | | 0.5% | 5.48 | 82.6% | |
| | | | 1% | 8.1 | 79.5% | |
| | | 0.1 | 0.1% | 2.12 | 69% | |
| | | | 0.5% | **6.89** | **84.7%** | |
| | | | 1% | 11.2 | 85.3% | |
| | Homogeneous | 0.02 | 0.2% | 0.53 | 67.0% | 88.6% |
| | | | 0.5% | 1.71 | 73.2% | |
| | | | 1% | 2.45 | 75.3% | |
| | | 0.04 | 0.2% | 1.5 | 73% | |
| | | | 0.5% | 3.06 | 84.1% | |
| | | | 1% | 5.10 | 85% | |
| | | 0.06 | 0.2% | 1.63 | 80.2% | |
| | | | 0.5% | 3.10 | 84.1% | |
| | | | 1% | **5.14** | **86.1%** | |

Table 16: The accuracy and example-level $\varepsilon$ of FedKT-L2 on Adult and cod-rna given different $\gamma$ values and number of queries. For each setting, we run 3 trials and report the median accuracy and the corresponding $\varepsilon$. The number of parties is set to 20 to ensure that FedKT has enough data to train each teacher model. The number of partitions is set to 1 and the number of subsets in each partition is set to 25. The failure probability $\delta$ is set to $10^{-5}$.

|  | Data Partition | $\gamma$ | #queries | $\varepsilon$ | acc | FedKT-L0 acc |
|---|---|---|---|---|---|---|
| Adult | Heterogeneous | 0.04 | 0.1% | 1.13 | 76.1% | 82.4% |
|  |  |  | 0.5% | 2.56 | 76.5% |  |
|  |  |  | 1% | 3.72 | 78.5% |  |
|  |  | 0.05 | 0.1% | 1.32 | 76.1% |  |
|  |  |  | 0.5% | 3.24 | 79.0% |  |
|  |  |  | 1% | 4.76 | 79.2% |  |
|  |  | 0.06 | 0.1% | 1.96 | 76.2% |  |
|  |  |  | 0.5% | 3.93 | 78.5% |  |
|  |  |  | 1% | **5.79** | **79.4%** |  |
|  | Homogeneous | 0.04 | 0.3% | 2.13 | 76.1% | 82.6% |
|  |  |  | 0.5% | 2.59 | 78.7% |  |
|  |  |  | 1% | **3.72** | **81.7%** |  |
|  |  | 0.06 | 0.3% | 2.97 | 76.3% |  |
|  |  |  | 0.5% | 3.93 | 79.9% |  |
|  |  |  | 1% | 5.79 | 81.8% |  |
|  |  | 0.08 | 0.3% | 3.77 | 76.3% |  |
|  |  |  | 0.5% | 5.04 | 80.4% |  |
|  |  |  | 1% | 7.89 | 82.0% |  |
| cod-rna | Heterogeneous | 0.04 | 0.5% | 3.54 | 77.7% | 89.7% |
|  |  |  | 1% | 5.14 | 79.8% |  |
|  |  |  | 2% | 7.63 | 82.0% |  |
|  |  | 0.05 | 0.5% | 4.51 | 81.4% |  |
|  |  |  | 1% | 6.58 | 82.0% |  |
|  |  |  | 2% | **9.78** | **84.7%** |  |
|  |  | 0.06 | 0.5% | 5.50 | 81.2% |  |
|  |  |  | 1% | 8.10 | 83.2% |  |
|  |  |  | 2% | 12.2 | 85.9% |  |
|  | Homogeneous | 0.03 | 0.5% | 2.64 | 79.2% | 90.6% |
|  |  |  | 1% | 3.78 | 80.5% |  |
|  |  |  | 2% | 5.51 | 83.1% |  |
|  |  | 0.04 | 0.5% | 3.54 | 80.1% |  |
|  |  |  | 1% | 5.14 | 83.7% |  |
|  |  |  | 1.5% | 6.45 | 84.0% |  |
|  |  | 0.05 | 0.5% | 4.51 | 80.8% |  |
|  |  |  | 1% | 6.58 | 84.2% |  |
|  |  |  | 1.5% | **8.3** | **84.7%** |  |

