# OpenReview forum: "Model-Agnostic Round-Optimal Federated Learning via Knowledge Transfer"
_ICLR.cc/2021/Conference — Reject_

### Official Review · AnonReviewer4 · 2020-10-28
**this paper considers a very interesting topic but lacks some important experimental comparisons and method is with limited novelty.**

**Rating:** 4
**Confidence:** 5

**Review:**

Paper summary:

This paper studies the  problem of  model-agnostic federated learning in the knowledge transfer framework.  They generalize the PATE framework in the federated learning setting by applying subsample-and-aggregate in each local agent. They provide theoretical analysis and empirical evaluation.

---------------------------------------

Merits:


The problem considered is indeed very important, as communication cost and privacy issue are two critical concerns in federated learning.

----------------------------
Concerns:

My main concern is the lack of novelty in their privacy analysis. The main theoretical part (Theorem 2 and Theorem 3) comes directly from PATE.

Moreover, the experiment part is insufficient. Noting most part of the paper focuses on differential private FedKT, I think the authors should provide at least one comparison with other privacy-preserving federated learning algorithms [1,2].

I have another concern about the feasibility of FedKT. FedKT requires splitting each local data into t splits and training a teacher model using each split. Noting a large t is well expected in the privacy setting, which may lead to poor performance on teacher models when local data is insufficient. One particular failure case is when the parties are mobile devices.

Regarding communication efficiency, I agree that FedKT is round-optimal. But I have doubts about the claim that FedKT's communication cost is lower than FedAvg.  The authors compare the overhead analysis with ```''nMs < 2nMr'' (the last paragraph on page 4) while ignoring the communication save from subsampling. FedAvg and its variants can benefit from subsampling clients (i.e., a subset of clients is sampled in each communication round). Moreover, in FedAvg and its DP variants [1,2], they choose a relatively large number of #parties while FedKT considers a small number of parties (#parties = 10). Note that this choice may degrade the performance of FedAvg in Table 1 as the variance across parties is larger. More could be done to establish that the experiments actually support the conclusions drawn about communication efficiency.


[1] Differentially Private Federated Learning: A Client Level Perspective.
[2] Learning Differentially Private Recurrent Language Models.

--------------------------------------
Overall this paper considers a very interesting topic but lacks some important experimental comparisons and method is with limited novelty.

---

> ### Author Response · Authors · 2020-11-25
> **Response to Reviewer 4**
>
> Thanks for your comments.
>
> **Q1**: Noting most part of the paper focuses on differential private FedKT, I think the authors should provide at least one comparison with other privacy-preserving federated learning algorithms [1,2].
>
> [1] Differentially Private Federated Learning: A Client Level Perspective.
>
> [2] Learning Differentially Private Recurrent Language Models.
>
> **A1**: [1] and [2] both only provide party-level differential privacy. FedKT provides both party-level and example-level differential privacy. Moreover, [1] and [2] cannot support tree-based algorithms (which are not trained by SGD) while FedKT does not have any requirement on the model structure. Since our current differential privacy experiments are conducted on the tree-based algorithm, [1] and [2] are not applicable. We will add experiments on MNIST/SVHN to compare with [1] for party-level differential privacy in the future.
>
> **Q2**: Noting a large t is well expected in the privacy setting, which may lead to poor performance on teacher models when local data is insufficient. One particular failure case is when the parties are mobile devices.
>
> **A2**: Please refer to A3 of the general response.
>
> **Q3**: The authors compare the overhead analysis with nMs < 2nMr (the last paragraph on page 4) while ignoring the communication save from subsampling.
>
> **A3**: Note that we focus on the cross-silo setting where the number of parties is relatively small as shown in the last two paragraphs of Page 1. Thus, FedAvg does not need to adopt the sampling technique. Moreover, the sampling technique will slow down the convergence compare with using all parties in each round. We have tried running FedAvg with sampling fraction 0.5 and it needs about 94 rounds to converge, while FedAvg without sampling needs about 42 rounds to converge. Although the sampling technique can reduce the communication cost per round, it increases the number of communication rounds.

---

### Official Review · AnonReviewer2 · 2020-10-29

**Rating:** 4
**Confidence:** 4

**Review:**

__Summary:__
The paper considers classification tasks in the federated learning scenario when each device/worker is powerful in terms of computational power and storage space, but, the communication between devices is constrained. The paper proposes a novel algorithm for federated learning that reduces the number of communication rounds to one. The algorithm constructs an ensembled model with the majority voting out of the locally trained models and then on the server side learns a final model by mimicking the performance of the ensembled model on a public dataset.

The paper additionally provides a differentially private version of the proposed algorithm and proves its privacy guarantees; and performs an experimental comparison of the proposed method.

__Main concerns:__
- The proposed approach is interesting and novel. However, I am not convinced that the setting is realistic. It is hard to imagine that devices with relatively high computational power are restricted so much on the communication side that cannot allow for more than 1 round.
- the algorithm does not achieve the best accuracy.
- The paper in general is hard to read.
- there is no theoretical understanding of how good the algorithm works. I can imagine many cases where the proposed algorithm wouldn't perform well in practice (small size of the public dataset / small size of local data on the workers / highly non-iid data on the workers)
- the algorithm is quite complicated and it is unclear if all the components are required. E.g. I don't see why is it needed to do ensembling of models locally, but not to have one local model trained on the full local data. This applies to the other algorithm's components as well.

__Ohter concerns:__
- Some related work on federated learning with knowledge distillation is missing, e.g. [1], [2], [3].
- The results are not reproducible: the tuning details and hyperparameters are not stated in the experiments
- some references are missing: e.g. for consistent voting on page 4.
- what is the number of queries on page 8, section 5.3?

[1]: D. Li, "FedMD: Heterogenous Federated Learning via Model Distillation"

[2]: T. Lin et al, "Ensemble Distillation for Robust Model Fusion in Federated Learning"

[3]: H. Chang, "Cronus: Robust and heterogeneous collaborative learning with black-box knowledge transfer."

---

> ### Author Response · Authors · 2020-11-25
> **Response to Reviewer 2**
>
> Thanks for your comments.
>
> **Q1**: However, I am not convinced that the setting is realistic. It is hard to imagine that devices with relatively high computational power are restricted so much on the communication side that cannot allow for more than 1 round.
>
> **A1**: We want to highlight that the novelty of our work lies not only in reducing the amount of communication but also in a single round of communication, which enables model-only learning possible. Specifically, we believe that new applications can be enabled if we can achieve the optimal communication round. If only a single communication round is needed, the parties only need to send the models and do not need to have further communication with the server. As stated in Applications paragraph of Page 5, model market for federated setting is possible, where the parties only need to sell the models to the market and federated learning can be done after a model trade.
>
> **Q2**: The algorithm does not achieve the best accuracy.
>
> **A2**: Please see A2 of our general response.
>
> **Q3**: The paper in general is hard to read.
>
> **A3**: Can you provide more details (i.e., which part is hard to read) so that we can improve the writing? Thanks.
>
> **Q4**: There is no theoretical understanding of how good the algorithm works. I can imagine many cases where the proposed algorithm wouldn't perform well in practice (small size of the public dataset / small size of local data on the workers / highly non-iid data on the workers)
>
> **A4**: (1) small size of the public dataset: We have added experiments to show the performance of FedKT with a smaller size of the public dataset in the revised manuscript. Please refer to Appendix B.5 and Table 10 of the revised manuscript. The accuracy of FedKT is generally statble. (2) small size of local data on the workers: We have provided experiments varying the number of parties (thus also varying the size of local data) in Appendix B.4 (see Table 8 and Table 9). The performance of FedKT is still good with a larger number of parties. Since we focus on the cross-silo setting, we do not try a very large number of parties. (3) highly non-iid data on the workers: We have provided experiments varying the imbalance level. Please refer to Appendix B.3. The performance of FedKT is still good with a highly non-IID data partition (i.e., $\beta=0.1$).
>
>
> **Q5**: The algorithm is quite complicated and it is unclear if all the components are required. E.g. I don't see why is it needed to do ensembling of models locally, but not to have one local model trained on the full local data. This applies to the other algorithm's components as well.
>
> **A5**: Note that we focus on the cross-silo setting as stated in the last two paragraphs of Page 1. Since the number of parties is relatively small and the data size in each party is relatively large, our multiple data partitioning in the local parties can increase the model diversity and thus increase the performance. For the other components (i.e., consistent voting, number of partitioning), we have provided ablation studies in Appendix B.2.1 and Appendix B.6. The experiments show that the components can improve the performance of FedKT.
>
> **Q6**: The results are not reproducible: the tuning details and hyperparameters are not stated in the experiments
>
> **A6**: We have provided the details. Please see the second paragraph of Appendix B.1.
>
> **Q7**: some references are missing: e.g. for consistent voting on page 4.
>
> **A7**: We propose the consistent voting approach. To the best of our knowledge, we do not know any related reference. Can you state the references clearly? Thanks.

---

### Official Review · AnonReviewer1 · 2020-11-02
**Good privacy analysis but not enough novelty**

**Rating:** 4
**Confidence:** 4

**Review:**

This submission proposes a new federated learning framework based on knowledge transfer. Local dataset at each party are partitioned and each partition is used to train a teacher model. All teacher models at each party are used to train a student model using pseudo labels based on voting on public dataset. Student model from each party is then uploaded to server and used to train the final model based on voting on unlabeled data. Differential privacy analysis is conducted and experimental evaluations comparing to other mainstream federated learning methods are presented. The advantages of the proposed method include privacy preservation, lower communication traffic, as well as applicability to non-differentiable models. While the proposed framework is technically sound, the reviewer is not convinced by its technical contributions. The design of the framework is integration of existing technics such as PATE, and the mechanism of protecting privacy as well as reduction of communication traffic is also not new (see FedMD: Heterogenous Federated Learning via Model Distillation,  Neurips 2019 Workshop and Ensemble Distillation for Robust Model Fusion in Federated Learning, Neurips 2020). There is no clear advantage in the proposed method over these existing methods from the reviewer’s point of view. Plus the overall performance on benchmark dataset seems to be degraded compared to other mainstream methods like FedAvg. The reviewer would like the authors to explain and discuss the technical contributions of the submission and compare the proposed framework to these similar existing methods based on knowledge transfer.

---

> ### Author Response · Authors · 2020-11-25
> **Response to Reviewer 1**
>
> Thanks for your comments.
>
> **Q1**: There is no clear advantage in the proposed method over these existing methods from the reviewer’s point of view.
>
> **A1**: Please see A1 of our general response.
>
> **Q2**: The overall performance on benchmark dataset seems to be degraded compared to other mainstream methods like FedAvg.
>
> **A2**: Please see A2 of our general response.

---

### Official Review · AnonReviewer3 · 2020-11-02
**Knowledge transfer applied to federated learning**

**Rating:** 5
**Confidence:** 4

**Review:**

===========================Post rebuttal===========================

I thank the authors for their responses and additional experiments. I understand that the focus of the paper is the cross-silo setting, however one of the key questions of an empirical study is to identify the limitations of the proposed approach. Experiments in Tables 8 and 9 still consider a relatively small number of clients and do not provide empirical insights into when the proposed approach begins to degrade. For future revisions, I recommend an empirical exploration that helps the reader to understand the limitations of the proposed method.

==================================================================

This paper explores the idea of knowledge transfer applied in the federated learning setting. Authors also consider variations of their algorithm under different privacy constraints.

Both knowledge transfer and differential privacy mechanisms are borrowed from prior works, limiting the methodological contribution of the paper. However, it was interesting to see these techniques applied in the FL context. The applicability of the proposed method relies on each client having sufficient data to train multiple teacher models locally. It would be interesting to see a deeper empirical study of the local data size effect: for example by varying the number of clients in the MNIST/SVHN experiments (therefore reducing the size of the local datasets). I also recommend a quantitative comparison of the communication costs by plotting accuracies of FedKT and baselines against the number of bytes exchanged between clients and the server.

I would consider increasing my score provided additional empirical studies I suggested (especially the effect of the local data sizes).

---

> ### Author Response · Authors · 2020-11-25
> **Response to Reviewer 3**
>
> Thanks for your comments.
>
> **Q1**: It would be interesting to see a deeper empirical study of the local data size effect: for example by varying the number of clients in the MNIST/SVHN experiments (therefore reducing the size of the local datasets).
>
> **A1**: We have provided the experiments by varying the number of clients in Adult, cod-rna, and MNIST in Table 8 and Table 9. Note that we focus on the cross-silo setting as stated in the last two paragraphs of Page 1. Experiments on a very large number of parties are out of our scope.
>
> **Q2**: I also recommend a quantitative comparison of the communication costs by plotting accuracies of FedKT and baselines against the number of bytes exchanged between clients and the server.
>
> **A2**: We have added the figures as shown in Figure 2 (c) and (d) of the revised manuscript. The communication cost of FedKT is very small (5.4MB and 7.5MB on MNIST and SVHN, respectively).

---

### Author Response · Authors · 2020-11-25
**Response to common questions**

Thanks for the reviewers' comments. We have updated the manuscript and added new contents as required (marked as red). Here are the common concerns and our response.

**Q1**: Compare with existing approaches [1,2,3] using knowledge transfer (lack of novelty).

[1] FedMD: Heterogeneous Federated Learning via Model Distillation

[2] Ensemble Distillation for Robust Model Fusion in Federated Learning

[3] Cronus: Robust and heterogeneous collaborative learning with black-box knowledge transfer

**A1**: We have added Section 2.4 to discuss existing approaches using knowledge transfer. Specifically,

[1] has a different setting with FedKT. [1] requires a public labeled dataset and FedKT only requires a public unlabeled dataset. Public unlabeled datasets widely exist in reality such as images and texts, while a public labeled dataset is more challenging to obtain in the federated setting.

[2] is a contemporaneous work recently published in NeurIPS 2020. It is not reasonable to ask for a comparison with [2] since the accepted list of NeurIPS 2020 is even unknown at the ICLR 2021 submission deadline. Also, please refer to FAQ section in Reviewer Guide (https://iclr.cc/Conferences/2021/ReviewerGuide). If a paper was published on or after Aug 2, 2020, authors are not required to compare their own work to that paper. Nevertheless, our work is different from [2], as FedKT is designed with a round-optimal goal.

[3] has a different objective with FedKT. [3] designs a robust federated learning algorithm to protect against poisoning attacks. Since it has a lower accuracy than FedAvg, it is not necessary to compare FedKT with [3].

In summary, all existing approaches transfer the prediction vectors (i.g., logits) on the public dataset between clients and the server. FedKT transfers the voting counts and thus can easily satisfy differential privacy guarantees with a tight theoretical bound on the privacy loss. Moreover, FedKT is designed with a round-optimal goal, while the other approaches use iterative learning algorithms that needs many communication rounds to converge. We have added a paragraph at the end of Section 1 to demonstrate our contributions.

**Q2**: The algorithm does not achieve the best accuracy.

**A2**: Our approach only needs a single communication round while the other approaches need multiple rounds to converge. As we have shown in Table 1 and Figure 2, our approach is much more communication-efficient than the other approaches. The accuracy of our approach is still comparable to the final accuracy of the other approaches. Moreover, as shown in Figure 2, we can achieve the best accuracy by using FedKT as the initialization step (see curve FedKT-Prox).

**Q3**: FedKT may not be applicable when the number of parties is large.

**A3**: As we have highlighted in the last two paragraphs of Page 1, we focus on the cross-silo setting [4], where the number of parties is relatively small and the parties have a relatively large computation power compared with the cross-device setting. Our 2-tier knowledge transfer design is based on the cross-silo setting. When it comes to the cross-device setting, the design can be totally different (e.g., see [5]) and is out of the scope of our paper. Moreover, we have provided experiments to vary the number of parties as shown in Appendix B.4. The performance of FedKT is stable in the range of our setting (see Table 8 and Table 9).


[4] Advances and open problem in federated learning

[5] Group Knowledge Transfer: Federated Learning of Large CNNs at the Edge

---

### Comment · ~Adam_Dziedzic1 · 2021-03-29
**PATE**

This is interesting work.

Better noisy aggregation mechanisms for teacher ensembles that are more selective and add less noise, and prove their tighter differential-privacy guarantees can be found in the "2nd PATE" paper: https://openreview.net/forum?id=rkZB1XbRZ (e.g., they used Gaussian in lieu of the Laplacian noise).

I think that it would be insightful to compare FedKT directly with PATE not only in terms of accuracy but also in terms of the privacy budget incurred for at least MNIST and SVHN datasets.

I was wondering why you create many student models per a single party P_i: "parties send their student models to the server". I'd expect a single student model per party.

CelebA and Chest X-ray datasets are for the multi-label classification but I see them used only for a binary classification task. You conduct a gender recognition on CelebA and predict only pneumonia on chest X-rays (Appendix B.7).

Other related papers: https://openreview.net/forum?id=h2EbJ4_wMVq and https://openreview.net/forum?id=NNd0J677PN

---

> ### Author Response · Authors · 2021-03-30
> **Response**
>
> Hi Adam,
>
> Thanks for your comments! We cited "2nd PATE" and pointed out a tighter DP guarantee by it (last paragraph of Section 4). We have noted the related papers and will add them in a future revision. We will consider adding the mentioned experiments if appropriate. For your question, creating many student models per party can improve the accuracy in the non-private setting (see Table 5). Thanks again!

---

### Decision · Program_Chairs · 2021-01-07
**Final Decision**

**Decision:**

Reject

**Comment:**

This paper proposes a model-agnostic FL method called FedKT that performs only one communication round and reduces the communication complexity of federated learning. The reviewers have the following concerns about the paper:
* Limited novelty because the proposed method is directly based on PATE
* Insufficient experiments

The authors did a great job of responding to the reviewers' comments and also added some new experimental results in the updated version. But the reviewers still recommend significant revision of the paper and resubmission to a future venue. I hope the authors will find their constructive and detailed comments below helpful!